# Molecularly-Imprinted SERS: A Potential Method for Bioanalysis

Hilda Aprilia Wisnuwardhani [1,2] , Slamet Ibrahim [1], Rino R. Mukti [3] and Sophi Damayanti [1,4,*]

1 Pharmacochemistry Research Group, School of Pharmacy, Institut Teknologi Bandung, Bandung 40132, Indonesia
2 Pharmacy Study Program, Faculty of Mathematics and Natural Sciences, Universitas Islam Bandung, Bandung 40116, Indonesia
3 Division on Inorganic and Physical Chemistry, Faculty of Mathematics and Natural Sciences, Institut Teknologi Bandung, Bandung 40132, Indonesia
4 University Center of Excellence on Artificial Intelligence for Vision, Natural Languange Processing & Big Data Analysis (U-CoE AI-VLB), Institut Teknologi Bandung, Bandung 40132, Indonesia
* Correspondence: sophi.damayanti@fa.itb.ac.id; Tel.: +62-813-9407-5730

**Abstract:** The most challenging step in developing bioanalytical methods is finding the best sample preparation method. The matrix interference effect of biological sample become a reason of that. Molecularly imprinted SERS become a potential analytical method to be developed to answer this challenge. In this article, we review recent progress in MIP SERS application particularly in bioanalysis. Begin with the explanation about molecular imprinting technique and component, SERS principle, the combination of MIP SERS, and follow by various application of MIP SERS for analysis. Finally, the conclusion and future perspective were also discussed.

**Keywords:** molecularly imprinted polymer; surface-enhanced Raman spectroscopy; bioanalysis





## 1. Introduction

Bioanalysis is related to drug development, forensic analysis, doping control, and identification of biomarkers for diagnostic methods of various diseases. Bioanalysis also provides information regarding the toxicokinetics, pharmacokinetics and pharmacodynamics of new drugs. Bioanalysis is the analysis of analytes, i.e., drugs, metabolites, and biomarkers, in biological samples (blood, plasma, serum, saliva, urine, feces, skin, hair, and organ tissues). Bioanalysis consists of several stages which include sample collection from preclinical and clinical trials, sample preparation, and bioanalysis stages using certain methods and instruments. The sample preparation stage is the most important stage in a bioanalysis. The role of the sample preparation stage is to remove the influence of the sample matrix, as well as to improve the analytical performance of a bioanalytical method [1].

Sample preparation helps increase the selectivity and sensitivity of bioanalytical methods. Due to the matrix complexity of biological sample, a sample preparation step is required [2,3]. Liquid-liquid extraction (LLE) and solid-phase extraction (SPE) are two commonly used techniques [1]. Liquid-liquid extraction has limitations, including minimal enrichment factor, inadequate recoveries, and requires large amounts of organic solvents [4]. Solid-phase extraction is usually used for cleaning and pre-concentration in analyzing biological samples due to the simplicity, rapidness, and minimizing of those limitations of LLE [4,5]. The main drawback of SPE is the selectivity of the sorbent that separates the analyte [5]. The use of SPE in sample preparation, is strongly influenced by selecting the suitable SPE sorbent. A good separation needed a selective and specific sorbent [6]. The SPE sorbent is the determining factor in the ultimate performance of the sample preparation procedure [7]. The most common adsorbent ($C_8$, $C_{18}$, $Al_2O_3$, silica)

are usually interfered by the sample impurities [8–10]. Molecularly imprinting technology allows us to create materials that can identify specific molecules to be analyzed and offer high selectivity of bioanalytical method [4]. Molecularly imprinted polymer (MIP) is a materials that mimic antigen-antibody reactions, so it can gained the specific recognition of analytes in sample preparation [11,12]. Because of that, MIP is widely employed in solid-phase extraction as a sorbent [3,4,10,13–24]. The use of MIP as a sorbent in SPE, can improve the selectivity, sensitivity and accuracy of the bioanalytical method.

Sample preparation is usually followed by detection using analytical instruments, such as HPLC, LC/MS, LC/MS/MS, SERS, electrochemical method, etc. Surface-enhanced Raman spectroscopy is a non-destructive, fast and sensitive method, particularly for trace analysis. The use of SERS depends on enhancement of Raman signal. Signal enhancement in SERS application, can happen if the analytes are adsorbed on rugged metal surfaces (e.g., Au, Ag, and Cu NPs) [25]. SERS has been utilized to detect trace organic chemicals, because of its facile procedure. The matrix interference effect, which includes non-targeted analytes and rugged Raman-like peaks from other molecules such as proteins, lipids, and pigments, prohibits it from being widely employed in complex matrix. The interference effect diminishes analyte sensitivity and can result in Au or Ag nanoparticle colloid precipitates. The matrix complexity of bioanalytical and trace analysis sample, can cause the failure in the enhancement Raman signals of SERS applications. These factors cause an increase in weak signal, matrix effect, and fluorescence interference from the background. These limitations can be overcome by the use of appropriate preparation techniques [26].

The combination of MIP and SERS as a sample preparation and detection method is one of the solution for bioanalysis. matrix MIP as a "smart" material that can increase the selectivity of sample preparation, was combined with SERS substrates (Au, Ag, CuNPs), into a new material, which can overcome the existing limitation [27,28].

Currently, published research article related to MIP-SERS are increasing time by time (Figure 1). This review discuss about current research regarding MIP-SERS application, particularly in bioanalysis. This review's scope is research articles published in the period 2016 to mid-2022. Compared to previous review articles (https://doi.org/10.1016/j.talanta.2020.122031 and https://doi.org/10.1021/acssensors.9b02039), this review article discusses more about the research related to MIP-SERS which were also published in 2021 and some from initial year of 2022. In addition, this review article discusses about MIP-SPE and SERS. The research articles presented in this review also discuss the variety of materials, especially combination of MIP and metal nanoparticles, which is used as SERS substrates.

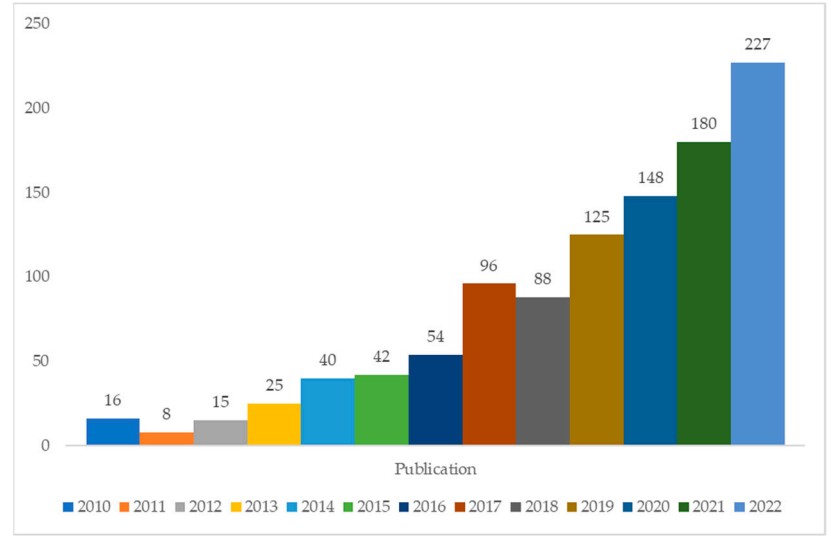

**Figure 1.** Publication with keywords of molecularly imprinted polymer, surface-enhanced Raman spectroscopy (according to Sciencedirect). Data was taken from August 2022.

## 2. Molecularly Imprinted Polymer (MIP)

MIPs (molecularly imprinted polymers) are synthetically generated high-affinity recognition materials. They are commonly utilized as a solid-phase extraction (SPE) sorbent because they efficiently separate and enrich analytes, particularly in complex matrix. Because of its recognition ability, chemical and thermal durability, easy synthesis, and low manufacturing cost, MIPs appear more promising than antibodies and aptamers [27,29–31].

Components needed for MIP synthesis are functional monomer, crosslinkers, porogen solvent, initiator, and template [30]. The molecularly imprinted polymer can be synthesized through covalent [24,32–34] and noncovalent methods [35–41]. Covalent imprinting procedures used reverse condensation processes such as Schiffs base, boronate ester, ketal, and acetal. Covalently prepared MIP's use covalent bonds to bind the target molecule to the functional monomer before polymerization, and the bond must be cleaved before the use of the MIP. Noncovalent imprinting techniques include hydrogen bonds, ion-pairs, dipole-dipole interactions, and van der Waals interactions. Nosncovalent polymer allows for easy removal of the target molecule and reversible binding during later use of MIP [42,43]. The methodologies for synthesis rely on the copolymerization of functional monomers and crosslinkers in the presence of a template or target molecule. The orientations and locations of the functional residue monomers are trapped in the polymer after polymerization, simulating a lock and key between a polymer and a target molecule [28].

### 2.1. Advantages and Limitations of MIP

The advantage of using MIP in the analysis process is the increased selectivity of the analytical method. MIP can extract analytes from samples with greater efficiency as it involves using molds suitable for the analyte. The molding process produces an active polymer site in a cavity that remains following a particular conformation. Affinity can also be maintained in the presence of hydrogen bonds. This causes the analyte to be easily captured at the active site of the MIP. MIP is flexible so that it can be used for various analytical purposes. MIP can also maintain stability and is sturdy in a broad pH and pressure range [44].

MIP's main drawback is template leakage [21,29,35,45–52]. This happens when not all templates are released during the template removal process, some are still left in the MIP cavity. To overcome this limitation, the utilization of dummy templates or templates analogous, can be chosen. Dummy template can be in the form of derivative compounds, those in the same group to analyte or the use of deuterated molecules or their isotope analogs. The use of this dummy template has proven to be able to overcome the shortcomings of MIP. The use of dummy templates can also overcome another drawback of MIP related to the availability of templates, where some templates are challenging to obtain due to price issues [45,48–52].

### 2.2. Component of MIP

The main components for MIP manufacture are templates, functional monomers, and crosslinkers. These components are needed, especially at the pre-polymerization stage, which is essential in manufacturing MIP.

#### 2.2.1. Functional Monomer

The functional monomer in the imprinted cavities supplies the functional groups that are important for the interactions involving the target molecule. The stronger the contacts during imprinting, the higher MIP's binding capacity and selectivity. Under rare circumstances, complex formation with the template molecule might influence monomer reactivity. A wide range of functional monomers with varying functionalities are commercially available. Figure 2 shows functional monomers used in noncovalent imprinting. Noncovalent MIPs are typically common method. The functional groups on the monomers will complement to a specific compound or class of compounds. Thus, basic functional monomers are chosen for templates containing acid groups and vice versa. Amphiphilic monomers can

be used to imprint low polar to nonpolar templates, resulting in hydrophobic or van der Waals forces hold monomer-template assemblies together [53,54].

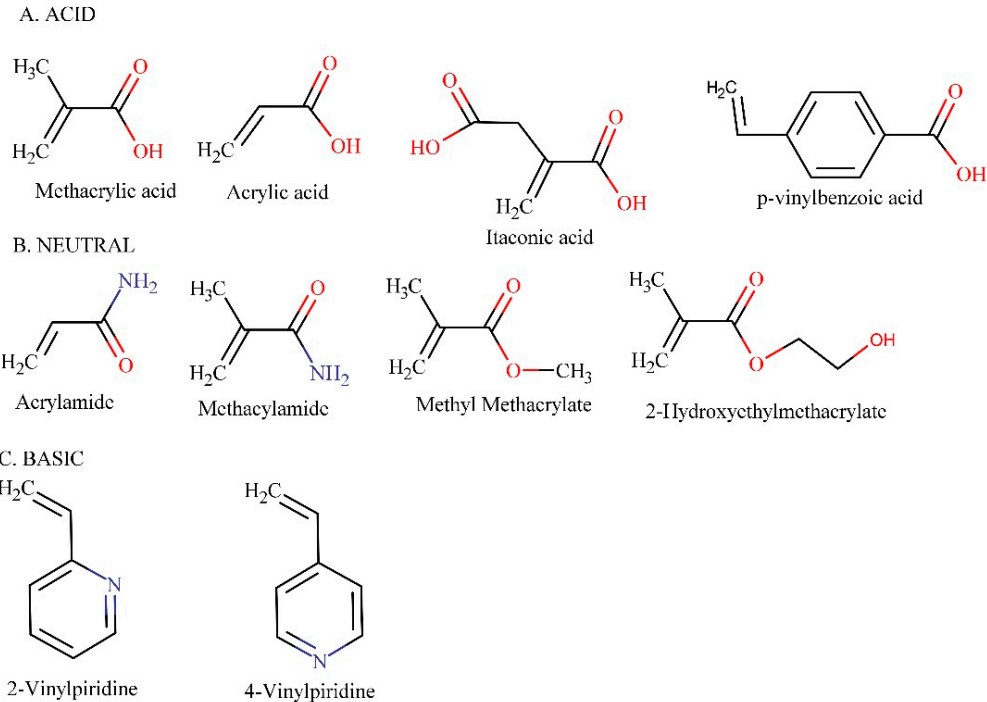

**Figure 2.** Functional Monomer.

Imprinting using a multiple functional monomers has also been revealed. Rather than the functional monomers' self-interaction, this technique requires the production of more durable interaction between the template and the functional monomers. In terms of recognition and selectivity, several of these materials out-perform the similar MIP produced with a lone functional monomer. The reactivity of the monomers should be matched to achieve copolymerization. The MIP's specificity is determined by the cavities form and size which is formed by the crosslinker, as well as the chemical interactions between the template and the functional monomer. As a result, choosing the functional monomers employed for MIP synthesis is critical to obtaining excellent results. Noncovalent interactions like as hydrogen bonds, dipole-dipole, ionic or hydrophobic interactions, are widely employed to generate MIPs. Therefore, a functional monomer is chosen according to the functional groups contained in the chemical structure of the template [55,56].

2.2.2. Templates

A template is a term given to the compound to be molded in MIP. The template is also usually the compound to be analyzed to obtain high specificity in analytical method. However, in its development, dummy templates or a combination of several templates can also be used. Each of its uses has a different purpose to MIP synthesis. The use of dummy templates is associated with template leakage from MIP particles during the sample preparation procedure [8,29]. In addition, some disadvantages of MIP are diffusion resistance, hard eluting, low binding rate, and deeply embedded template in the internal also predicted to be overcome by using this dummy template strategy [57]. This limitation can affect the results of the analysis. This solution can result in better analyte recognition capability for MIP. Dummy template might be a material with a similar chemical structure to the analyte to be isolated. For example, the use of 2-chlorophenothiazine in MIP production for phenothiazines analysis in meat samples. Because they both have a thiodiphenylamine ring in their chemical structure, 2-chlorophenothiazine was chosen [9]. MIP was also created using dummy templates for the detection hordenine analysis in urine

samples [4], fluoroquinolones and sulfonamides in pig and poultry samples [8], benzimida-zole analysis [29], caffeine analysis in wastewater samples [50], morphine analysis in urine samples [57], polybrominated diphenyl ethers [58], bisphenol A analysis [59], ractopramine analysis [60], and acrylamide analysis [61].

Multiple templates can be used to create MIPs. It is designed to analyze specific groups of compounds, and this multi-template selection is more concerned with achieving MIP selectivity for a specific class of compounds. The use of multi-templates in the manufacture of MIP includes ibuprofen, naproxen, and diclofenac for the analysis of acidic active pharmaceutical compounds. The results demonstrated that the multi-template MIP has good molecular recognition properties because it can simultaneously extract all three target compounds. In contrast, the single-template MIP can only extract one analyte [62].

Multiple templates are used in the preparation of MIP to detect nitrosamines in water and beverage samples. The five templates employed are nitrosamine chemical derivatives observed in water and beverage samples. As indicated by strong adsorption capacity values and good selectivity for the five chemicals, the MIP created could evaluate five nonpolar nitrosamine derivatives [30].

### 2.2.3. Crosslinkers

The crosslinker grips the functional groups within the selective binding sites for template recognition by stabilizing the imprinted cavities. The kind and amount of crosslinker utilized also impact the shape and stability of the structure. A high crosslinker to functional monomer ratio results in more stiff materials with reduced swelling capacities since the structure cannot expand. Polymers created with low molecular weight crosslinkers have higher stiffness than those prepared with another one. It creates polymers with higher quality selectivity, affinity, and binding capacity. As a result, the kind and extent of crosslinker utilized in the imprinting process must be selected carefully [63–69].

The ratio of crosslinkers utilized in relation to the total number of moles of functional monomers is relatively high. A mechanically strong polymer with a persistent porous structure and a large surface area was developed at this concentration. A low number of crosslinkers results in sticky polymers with restricted imprinting applications. For copolymerization, the reactivity of the functional monomers and the crosslinker must be matched. Figure 3 depicts a range of crosslinkers often employed in molecular imprinting.

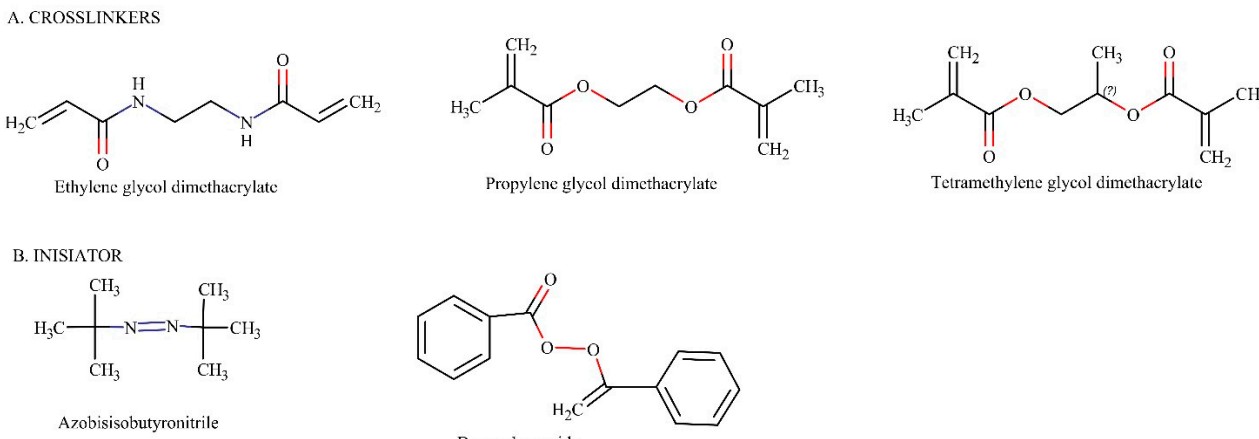

**Figure 3.** Crosslinkers and Inisiator.

### 2.2.4. Initiator

MIP is typically produced by free radical polymerization (FRP) either thermally or photochemically generated. This process has three stage i.e., initiation, propagation, and termination. The rate of polymerization increases during the early phases of radical breakdown. Azo compounds, peroxo, redox systems, and photoinitiators are some of the most often utilized initiators in free radical polymerization. Azo initiators can produce free

radicals when exposed to UV light at maximal wavelengths or when heated. The most utilized azo initiator in MIP production at low polymerization temperatures, precisely 60 °C and 40 °C, is 2,2′-azobis(isobutyronitrile) (AIBN) [37,70–72]. Figure 3 depicts a typical initiator used in molecular imprinting.

### 2.3. Rational Study of MIP Synthesis

Computational studies of MIP include molecular mechanics, quantum mechanics (ab initio, semiempirical, and density functional theory), and molecular dynamics. It is best explained using the Ab initio method. It explains a system's electronic structure better to explain noncovalent interactions between templates and monomers. However, the decision should consider processing costs, calculation accuracy, and the amount of compatibility between theoretical calculations and practical performance. The results obtained using the ab initio approach will be more accurate, but the time required will be longer. Several studies use the ab-initio method with the Hartree-Fock method basis set HF/6–31+G** [2], HF/3–21G [73], HF/6–31G(d) [74] for calculations. Some use a combination of RHF and DFT methods [2,75] or HF and DFT methods [76].

The semiempirical method will calculate the bond energy faster. The most widely chosen approaches are AM1 (small atomic data) and PM3 (large molecular properties) [6]. The accuracy of the semiempirical method depends on the parameters available for the target molecule. The results will be good if the target molecule has been contained in the database. The density functional theory (DFT) method is preferred because it produces balanced results between cost and accuracy. Computational studies have preceded many studies to shorten the laboratory optimization time. Several studies using a semiempirical approach are serotonin analysis using the PM6-DH2 method, followed by calculations utilizing DFT approach using the B97XD/6–31++G (d,p) method [77]; PM3 methods for MIP manufacturing of domoic acid enrichment from seawater and shellfish [78] and for manufacturing MIP for erythromycin detection based electrochemical sensor [79]. In addition to molecular mechanics and quantum mechanics, molecular dynamics studies can also be used for MIP computational studies. Molecular dynamics studies can simultaneously simulate the effect of time on interacting atomic groups computationally.

The most used computational approach for rational MIP synthesis is density functional theory. This is since DFT can bring benefits in terms of accuracy and cost. To generate selective MIP, hybrid DFT approaches like as B3LYP are frequently utilized to determine the binding energy between templates and functional monomers. The B3LYP method is widely applied in various variations of the basis set, including B3LYP/6–31G(d,p) [30,70,80–86], B3LYP/6–31G+(d,p) [4,9,59,81,82], B3LYP/6–311G [27,87,88] B3LYP/6–31+G(2d,2p) [11], B3LYP/6–311+G* [89], B3LYP/Aug-cc-pVDZ [36], B3LYP/6–311+G (d,p) [90]. The basis set is extensively used to find the optimum functional monomer and to compare the template to functional monomer.

The B3LYP functional calculates the relative contributions of the different component exchange and correlation terms using parameters. One disadvantage of the B3LYP technique is that it cannot reliably anticipate physical dispersion. Meanwhile, the DFT method should predict the complete repulsive interaction in a dispersion bond system. Other methods, such as M05, M05-2X, M06, M06-2X, and M07, M07-2X, perform reasonably well for binding energies of non-covalently bonded dimers like those in the fit set [91]. Therefore, many studies in recent years have used the M06-2X method as a better alternative than B3LYP [20,92–94].

MD simulations are used to rationally design a molecularly imprinted system and evaluate the molecular level process. This approach has the potential to considerably increase the efficiency of creating molecularly imprinted materials, lowering material costs, and minimizing experimental time. Several research have also employed molecular dynamics approaches in MIP synthesis computational experiments. Few research, however, have concentrated on the link between the template and the optimal monomer. The MD

study gives information on the mechanics of template recognition in these molecularly imprinted materials.

An example is the research of Shoravi et al. [95] conducting a molecular dynamics test using the AMBER®, by first preparing the MIP pre-polymerization mixture. Molecular dynamics tests were carried out to determine the best pre-polymerization composition for the manufacture of oseltamivir MIP. Another study conducted by Kong et al. [96] used different software and force fields to determine template (norfloxacin) interaction mechanism with functional monomers. Madikizela et al. [64] also used the same software and force fields to determine the intermolecular interactions of templates with functional monomers. This method is used for computational tests on MIP manufacture for acidic pharmaceutically active compounds using multiple templates. Bates et al. [97] perform a molecular dynamics study on MIP's manufacture to analyze melamine in milk samples.

*2.4. MIP Application in Analytical Chemistry*

MIP has been widely used for chemical analysis and drug delivery in the pharmaceutical industry. In analytical chemistry, MIP is used particularly for sample preparation. In chemical analysis, sample preparation is critical. The application of MIP can improve the analytical method's selectivity. This is due to the MIP cavity's ability to preferentially attach to the same or analogous analytes as the template utilized during the fabrication process. As a result, MIP is commonly utilized as a sorbent in solid phase extraction. This is due to the use of less selective sorbents in solid-phase extraction. MIP may also be used as a stationary phase in chromatographic separations, which is a sort of chemical analysis. It may also be utilized as a chemical and biological sensor and probe.

MIP's application as a sorbent is not confined to its usage as a sorbent for solid phase extraction (SPE). dSPE (dispersive solid phase extraction), MSPE (magnetic solid phase extraction), and SPME are further separation techniques that use MIP as a sorbent (solid phase microextraction) [98,99]. The results of sample preparation using MIP are usually followed by detection using various methods. Some of the analytical methods that are often used are spectrophotometry [98–100], spectrofluorometry [101–104], liquid chromatography [105,106], gas chromatography [107], Raman spectroscopy, electrophoresis, electrochemical [108–116] methods.

## 3. Surface-Enhanced Raman Spectroscopy

Raman spectroscopy was invented in 1930. However, its use is restricted due to its low sensitivity and weak Raman signal intensity. Along with the invention of the LASER and Van Duyne's study, which demonstrated an increase in signal related to the silver electrode's surface roughness induced by adsorption of tiny molecules, it developed a signal-enhancing phenomenon known as surface-enhanced Raman scattering. Because SERS has a high sensitivity (up to $10^4$), it offers great promise for application in studying and detecting single molecules [25,117].

Gold (Au), silver (Ag), and copper (Cu) are usually utilized as SERS substrates. They are stable, give strong SERS signal enhancement, and are inexpensive. Because the size of metal particles influences its efficacy as a substrate and SERS signal enhancer, those metals are often utilized in the form of nanoparticles. The distance between metal nanoparticles and the analyte (also known as *hotspots*) improves the SERS signal. Salt can also trigger nanoparticle agglomeration, increasing hotspots and resulting in stronger SERS signals [25,118–120].

SERS signal enhancement may be explained by two mechanisms: electromagnetic enhancement (EM) and chemical enhancement (CE). SERS offers various benefits in analysis, including high detection sensitivity, rapid signal creation if the target molecule is adsorbed on the surface of the SERS-active material, and fingerprint features [121] that can validate the chemical structure based on energy levels. SERS has applications in agricultural chemicals [26,117], adulteration [117], biological toxins found in agricultural goods [117], clinical diagnostics [118], veterinary drugs, food contaminants [119], environmental pollutants [120,121], and biology [122,123].

In the presence of a complex matrix, the SERS signal might be disturbed. Interfering substances can create false-negative signal readings or fail to identify the investigated component in complex matrix. On the other hand, interfering chemicals can result in erroneous positive signal readings. As a result, SERS was combined with other approaches to limit the influence of matrix interfering chemicals. Furthermore, MIP may be utilized as a material in the adsorption process and to eliminate interfering chemicals from the matrix (capturing the SERS substrate to get closer to the target molecule) [27,123,124].

## 4. Molecularly-Imprinted SERS Methods

MIP-SERS combination approach may be used to detect compounds precisely and sensitively. It can be done by one or two-step MIP-SERS. The SERS substrate is adsorbed onto the MIP surface in one-step MIP-SERS, allowing separation and detection to be completed in a single step. While in two-step MIP-SERS, the separation of analyte is distincted with the detection. The distance between the SERS substrate and MIP with the target molecule is crucial in the one-step MIP-SERS. One-step MIP-SERS are classified into core-shell, planar, and sandwich [117]. Some examples of the MIP-SERS application scheme can be seen in Figure 4. Table 1 shows some of the MIP-SERS applications in analysis.

### 4.1. One Step MIP-SERS

Ren, et.al, developed benzimidazole analysis using the single-step MIP-SERS method. The type used is the core-shell formation type. Ag microspheres were used as the core, and then coated with MIP. MIP in this study was synthesized using a dummy template, namely carbendazim. The use of carbendazim as a dummy template and reducing background noise can also avoid the phenomenon of template leakage in MIP making. Based on the validation results of the analytical method, this method was proven to be used for qualitative and semi-quantitative analysis of benzimidazole on samples with complex matrix [28].

Another study developed an analytical method for detecting Rhodamine 6G using $SiO_2/Ag/MIP$ nanocomposites. $SiO_2$ is used as a buffer that will adsorb Ag+ ions, which are then reduced with ethanolamine. In this study, $SiO_2/Ag$, which acts as a substrate and acts as a core, was then covered with an MIP using a surface molecularly imprinting technology (SMIT) method. The combination of $SiO_2/Ag$ with a MIP is expected to overcome the shortcomings of MIP in the form of low binding capacity and low bond kinetics. MIP was synthesized by the precipitation polymerization method. The thickness of the MIP layer on the $SiO_2/Ag$ surface is regulated by adjusting the number of crosslinkers in the pre-polymerization reaction. The thickness of this layer will affect the detection with SERS. The ratio of monomer and crosslinker is 1:3, giving the maximum MIP layer thickness for SERS detection, which is 40 nm. The ratio increases, the SERS signal decreases [49].

Hu et al., also developed a nanocomposite between AgNPs and MIP (AgNPs@MIP) for the detection of caffeine residues in wastewater. This study used a dummy template, theophylline, which is similar to caffeine. In this study, AgNPs were spread on MIP, then the formed nanocomposite was used as an adsorbent for SPE cartridges. The one step MIP-SERS method of this type can reduce the shortcomings of the one step MIP SERS method with the core shell type which requires synthesis conditions that are difficult to control and reproduce. This method can detect caffeine in river water samples with a fast analysis time of 23 min. The same mechanism was used for the analysis of bisphenol A on polycarbonate plastic samples. AgNPs substrates are formed in situ in MIP. The AgNPs formed are expected to be evenly distributed to support the analyte-AgNPs interaction and increase the hotspot effect, which can increase the signal at the time of SERS detection [50].

**Table 1.** MIPSERS application in analysis.

| No. | Chemical/Biological Compounds | Samples | Methods | Noble Metal | Functional Monomer (FM) | Template | Crosslinker | Rational Study | Analytical Performance | Ref. |
|---|---|---|---|---|---|---|---|---|---|---|
| 1. | Bitertanol | Food | MIPSERS | Au | MAA (methacrylic acid) | Triamedifon (dummy template) | Trimethylolpropane trimethacrylate (TRIM) | ND | LOD Cucumber: 0.041 mg/kg Peach: 0.029 mg/kg | [27] |
| 2. | Benzimidazole | Preliminary study | MIPSERS | Ag | MAM | Carbendazime (dummy) | EDGMA | ND | LOD: $1.0 \times 10^{-8}$ mol/L | [28] |
| 3. | Caffeine | Wastewatere | MIPSERS | Ag | MAA | Theophylline (dummy) | EGDMA | ND | LOD: 100 ng/L | [50] |
| 4. | 2,6-dichlorophenol | Water | SGA MIP SERS | Au | MAA and AM | 2,6-dichlorophenol | EGDMA | ND | LOD: 200 nmol/L | [56] |
| 5. | Enrofloxacin hydrochloride | Water | AgMIM SERS | Ag | AM | Enrofloxacin hydrochloride | EGDMA | ND | LOD: $10^{-7}$ mol/L | [71] |
| 6. | Triazine fungicide | Rice and wheats | MIPSERS | Au | MAA | Prometryn and Simetryn | Trimethylopropane trimethacrylate (TRIM) | ND | Recoveries: 72.7–90.0% | [118] |
| 7. | Patulin | Fruits | MIPSERS | Au | 4-vinylpiridine (VP) | Patulin | 1,4-Diacryloylpiperazine (PDA) | ND | LOD: $5.67 \times 10^{-12}$ M | [119] |
| 8. | Bisphenol A | Tap water | MIPSERS | Ag | 4-vinylpiridine (VP) | Bisphenol A | EDGMA | ND | LOD: $1 \times 10^{-9}$ mol/L | [120] |
| 9. | Rhodamin 6G | Water | ZOAMIPSERS | Ag | AM (acrylamide) | Rhodamin 6G | Ethyleneglycol dimethacrylate (EDGMA) | ND | LOD: $10^{-13}$ mol/L | [121] |
| 10. | Carcinoembryonic antigen (CEA) | Serum | MIPSERS | Au | 4-vinylbenzeneboronic acid (VPBA) | Carcinoembryonic antigen (CEA) | EDGMA | ND | LOD: 0.1 ng/mL | [122] |
| 11. | λ -Cyhalotrin | Water | SGA MIP SERS | Ag | MAA and AM | Cyhalotrin | EDGMA | ND | LOD: $3.8 \times 10^{-10}$ mol/L | [124] |
| 12. | Paracetamol | Waste water | MIPSERS | Au | MAA | Paracetamol | EDGMA | ND | LOD: 300 nM | [125] |
| 13. | Carbamate pesticides | Tap water | MIPSERS | Ag | Methylacrylamide (MAM) | Carbaryl and thiodicarb | EDGMA | DFT B3LYP level basis set 6–31G(d) | Recoveries Carbaryl: 86.0–89.7% Thiodicarb: 79.0–84.7% | [126] |
| 14. | Sulfamethazine | Meat | Ag-TiO$_2$ MIP SERS | Ag | MAA and AM | Sulfamethazine | EDGMA | DFT to obtain molecular electrostatic potential (MEP) | LOD: $3.6 \times 10^{-9}$ mol/L | [127] |
| 15. | Histamine | Liquor, vinegar, prawn | MIPSERS | Ag | MAA | Histamine dihydrochloride | EDGMA | ND | LOD: $3.088 \times 10^{-9}$ mol/L | [128] |

**Table 1.** *Cont.*

| No. | Chemical/Biological Compounds | Samples | Methods | Noble Metal | Functional Monomer (FM) | Template | Crosslinker | Rational Study | Analytical Performance | Ref. |
|---|---|---|---|---|---|---|---|---|---|---|
| 16. | Tyrosine | Aqueous medium | PDA MIP SERS | Ag | AM | Tyrosine | EDGMA | ND | LOD: $10^{-9}$ mol/L | [129] |
| 17. | p-nitroaniline | Water | DG/Ag-MIP SERS | Ag | Methacrylamide | p-nitroaniline | N, N, N′, N′-Tetramethylethylenediamine (TEMED) | ND | LOD: $1.0 \times 10^{-14}$ M | [130] |
| 18. | Antibiotics | Water | Ag/ESM SERS | Ag | AM | Spiramycin | EGDMA | ND | LOD: 0.027 nmol/L | [131] |
| 19. | Metformin HCl and Phenformin HCl | Hypoglycemic health product | MIP@Au-GO SERS | Au | MAA | Metformin HCl | EGDMA | ND | LOD: 0.1 mg/mL | [132] |
| 20. | Malachite green | Fish muscles | Au@AgNPs MIP SERS | Au and Ag | MAA | Abietic acid (dummy template) | EGDMA | Optimization: DFT M06-2X/6–31G** Binding energy: Basis set def2TZVP with or without zero-point energy correction (ZPEC) | LOD: 0.37–0.64 ng/g | [133] |
| 21. | Malachite green | Water and carp | MIP@Fe$_3$O$_4$ SERS | Ag | MAA | Malachite green | EGDMA | ND | LOD: Tap water: 1.50 pM Carp: 1.62 pM LOQ Tap water: 4.96 pM Carp: 5.38 pM | [134] |
| 22. | Propranolol | Complex samples | GO-MIP SERS | Ag | MAA | Propranolol | EGDMA | ND | LOD: $10^{-11}$ mol/L | [135] |
| 23. | 2,4-dichlorophenoxyacetic acid | Milk | MISPE SERS | Ag | 4-VP | 2,4-dichlorophenoxyacetic acid | EDGMA | ND | LOD: 0.006 ppm LOQ: 0.008 ppm | [136] |
| 24. | Chlorpyrifos | Apple juice | MIPSERS | Ag | MAA | Chlorpyrifos | EGDMA | ND | PLSR RMSEC: 0.0453 RMSECV: 0.1470 | [137] |
| 25. | Thiabendazole | Orange juice | MISPE SERS | Ag | MAA | Thiabendazole | Divinylbenzene | ND | LOD: 4 ppm | [138] |
| 26. | Atrazine | Apple juice | MIP SERS | Au | MAA | Atrazine | EGDMA | ND | LOD: L-AuNPs: 0.005 mg/L–0.01 mg/L M-AuNPs: 0.01 mg/L–0.05 mg/L S-AuNPs: 0.01 mg/L–0.05 mg/L | [139] |
| 27. | L-Phenylalanine | Serum | Au@MIP SERS | Au | Phenyltrimethoxysilane (PTMOS) | L-Phenylalanine | Tetraethyl orthosilicate (TEOS) | ND | LOD: 1.0 nmol/L | [140] |
| 28. | Bisphenol A | Polycarbonate plastic | Ag@MIP SERS | Ag | MAA | Bisphenol A | EGDMA | ND | LOD: $5 \times 10^{-8}$ mol/L | [141] |

**Table 1.** *Cont.*

| No. | Chemical/Biological Compounds | Samples | Methods | Noble Metal | Functional Monomer (FM) | Template | Crosslinker | Rational Study | Analytical Performance | Ref. |
|---|---|---|---|---|---|---|---|---|---|---|
| 29. | Enrofloxacin hydrochloride | Water | Fe$_3$O$_4$@Ag@MIP SERS | Ag | Dopamine | Enrofloxacin hydrochloride | Dopamine | ND | LOD: 0.012 nmol/L | [142] |
| 30. | Enrofloxacin hydrochloride | Water | AGP MIM SERS | Ag | AM | Enrofloxacin hydrochloride | EGDMA | ND | LOD: 0.0078 nmol/L | [143] |
| 31. | Lysozyme | Clinical uses | AgMIP SERS | Ag | MAA and AM | Lysozyme | N,N-methylene acrylamide | DFT and MEP | LOD: 5 ng/mL | [144] |
| 32. | p-nitroaniline | Water | Ag@MIP SERS | Ag | Methylacrylamide | p-nitroaniline | EGDMA | ND | LOD: $10^{-12}$ M | [145] |
| 33. | PAH (polycyclic aromatic hydrocarbon) | Creek water and seawater | Au@MIP SERS | Au | MAA | Pyren and fluoranthene | Divinylvbenzene (DVB) | ND | LOD: 1 nM | [146] |
| 34. | Cloxacillin | Pig serum | MMIP SERS | ND | MAA | Cloxacillin | EGDMA | ND | LOD: 7.8 pmol | [147] |

ND: Not determined.

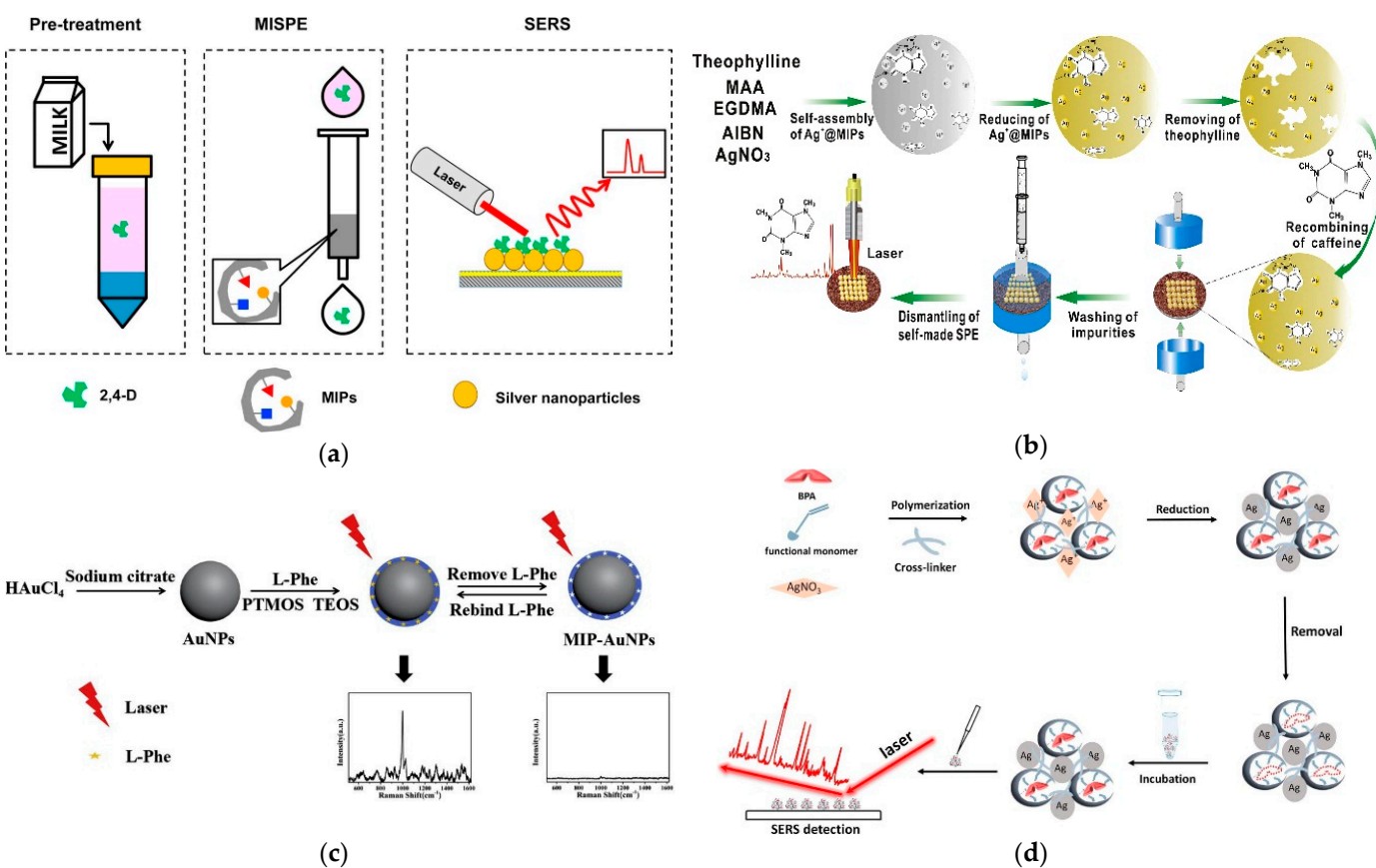

**Figure 4.** Example of MIP SERS scheme application for analysis (**a**). 2,4-dichlorophenylacetic acid; (**b**). caffeine; (**c**). L-Phenylalanine; and (**d**) Bisphenol A. Reuse with permission from [51,128,137,142].

Li et al., developed an analytical method for 2,6-dichlorophenol using $SiO_2/rGO/Au$ composites, SGA, as SERS substrate. The composites made are expected to increase the sensitivity of the SERS substrate. The combination of composites with MIP is further expected to increase the selectivity of the SERS substrate. Through the SMIT (surface molecularly imprinting technology) mechanism, it is hoped that specific cavities that recognize certain compounds can freeze on the surface of the SERS substrate. In addition, in the manufacture of MIP, a combination of two functional monomers, i.e., methacrylic acid and acrylamide, was used to increase the potential for template recognition to the formed cavity. This is expected to increase the selectivity and sensitivity of the MIP-SERS method. The use of SGA MIP SERS for the detection of 2,6-dichlorphenol in river water samples, showed a good recovery value (98.74–104.75%) and a linear range from 100–1.0 nmol/L [56].

Wu et al. conducted research to develop patulin analysis methods on fruit products. Patulin is a secondary metabolite produced by fungi that often contaminate fruit products. The developed method is a one-step MIP, in which AuNPs as a substrate is then coated with MIP, which is synthesized using patulin as a template. MIP synthesis was carried out using the free-radical polymerization method. The analysis results using the MIP-SERS method showed the same good results as the previous method (MIP coupled with quantum dots, MIP-EC, LC-MS, and conductometric methods). This method also provides good selectivity in the presence of analytical confounders such as OXD (oxindole) and 5-HMF (5-hydroxymethylfurfural) [119].

Bisphenol A analysis method was also developed using the MIP-SERS method. Ag@MIP synthesis begins with the synthesis of silver nanoparticles (AgNPs). MIP-SERS was carried out by a one-step method, where AgNPs as substrates acted as cores, while MIPs were superimposed on the surface of AgNPs and acted as shells. The formed AgNPs were then surface modified with the addition of APTES. Furthermore, MIP was synthesized by the

non-covalent method on the surface of the modified AgNPs. The mechanism of the one-step MIP-SERS analysis is thought to be the same as that of the Rhodamine 6G analysis, namely through the "gate effect" mechanism. The analysis results show an excellent detection limit value, but it has a drawback that not all BPA used in the MIP synthesis process can be released at the binding site of MIP. Therefore, in the subsequent development, it is hoped that dummy templates can be used to improve the performance of this method [120].

Li et al., developed the one-step MIP-SERS method. In this study, a combination of ZnO/Ag was used as a substrate, then MIP was coated on the nanocomposite surface for further use in the analysis of rhodamine 6G. This research involves a different mechanism with hotspots, where the substrate must be at a certain distance from the analyte to be analyzed. The mechanism that occurs in this study is the "gate effect", where the substrate coated with MIP can still detect the presence of the analyte, through a channel that connects the analyte to the substrate. The detection response produced by using ZnO/Ag as a substrate is better than using ZnO or Ag alone as a substrate [121].

Different mechanisms are shown by the analytical method developed by Feng et al. This analytical method was used to detect carcinoembryonic antigen (CEA) from serum. The single-stage MIP-SERS mechanism used is the sandwich type. This study used nanotags composed of AuNPs that have been modified with the addition of MPBA (4-mercaptophenylboronic acid) on the surface. The results show that further development is needed. From the analysis results using Raman spectroscopy, it is known that template leaks are still detected through the background of the spectrum. This causes a low value of the signal-to-noise ratio. Therefore, this method still requires further development [122].

The exact mechanism is also suspected to occur in using $SiO_2/GO/Ag$ nanocomposite as a substrate in the analysis of λ-cyhalothrin in water samples. $SiO_2/GO/Ag$ then acts as a core which will be coated on the surface by MIP. Previously, the nanocomposite surface was modified with polydopamine (pDA). MIP synthesis uses a com-bination of two functional monomers. The results showed that using $SiO_2/GO/Ag$ (SGA) nanocomposite as a sub-strate gave a better increase in SERS signal than using AgNPs alone. The addition of pDA on the nanocomposite surface also led to better substrate dispersion to increase the sensitivity and selectivity of this method [124]. Decorbie et al. using Au nanocylinder to analyze paracetamol residues in water samples. By using the one-step MIP-SERS method, it is known that the analytical method provides good sensitivity and selectivity and can be used for routine analysis [125].

The one-step MIP-SERS method with core-shell type was applied by Cheshari et al., for the analysis of pesticide residues (carbaryl and thiodicarb) in agricultural products. The uniqueness of this research is to compare the use of a single template and dual templates in MIP synthesis. This research also uses a computational approach to predict intermolecular interactions between templates and monomers using the molecular electrostatic potential (MEP) method. The results showed that the results were synchronous between the computational approach and those carried out in the laboratory. The use of dual templates in making MIP results in better selectivity than single templates [126].

Ren, et.al, used $Ag@TiO_2$ composite as a substrate in one-step MIP-SERS analysis for sulfamethazine compounds. $Ag@TiO_2$ is predicted to have the ability to clean the remnants of the template left by a photolytic mechanism. There were still about 5% of the template which was challenging to clean from the surface of the MIP cavity. Results showed that $Ag-TiO_2@MIP$ could be used for routine analysis in the laboratory [127].

Chen et al., used a nanocomposite which is a combination of two semiconductors (ZnO and $TiO_2$) with Ag as a substrate in one-step MIPSERS analysis. This nanocomposite is expected to improve the detection of SERS signals during analysis. The type used is the core-shell type. The results showed that $ZnO@TiO_2@Ag$ nanocomposite can be used as a substrate and has good sensitivity, selectivity, and accuracy for histamine analysis in food products [128].

Hi et al., developed a tyrosine analysis method using SERS. The substrate was made in a composite between PVDF/pDA/Ag. PVDF (polyvinylidene fluoride) was chosen

because it has a rough surface, so it is expected to increase the "hotspot" effect which plays a role in increasing the SERS signal. The PVDF/pDA/Ag composite formed was then surface modified with vinyl from methacryloxypropyl tri-methoxy silane (MPS). MIP is then printed to form a PDA/MIM on the modified surface. In this study, MIP was made with a two-stage precipitation polymerization reaction. This tyrosine detection method uses a sandwich type. The results show that this analytical method has a good recovery percentage, and the detection limit is equivalent to the previous method ($10^{-9}$ mol/L), but with a shorter analysis time of only 1.0 min [129].

Analysis of p-nitroaniline in water samples can also be carried out using the MIP-SERS combination method. Ag substrate was made in the form of nanocomposite with graphene, then MIP was copolymerized on the surface of DG/Ag using p-nitroaniline as a template. The use of graphene as a supporting material because of its 2-dimensional morphology provides a large surface area, making it suitable for use in the sample preparation stage. Graphene morphology can protect Ag from oxidation. Graphene material can also increase the signal of Raman spectroscopy through chemical enhancement mechanism. The results showed that the combination of the MIP SERS method could be used for the analysis of p-nitroanaline in environmental samples [130].

Wang et al., combined the technology of imprinting, membrane separation and detection with SERS for the analysis of enrofloxacin in water samples. A poly(vinylidene fluoride) (PVDF) membrane was used as a buffer. AgNPs are then dispersed on the membrane surface. MIP is then superimposed on the surface of the support and AgNPs, thus forming a sandwich-like shape. The interaction between enrofloxacin and the substrate was estimated based on the "gate" effect. It was suspected that on the surface of the imprinting layer there was a channel that could connect the substrate with the compound to be analyzed. Therefore, the detection of SERS for enrofloxacin in this study was thought based on an electromagnetic enhancement mechanism [71]. MIP SERS application for antibiotic residue analysis was also developed using $Fe_3O_4$/Ag nanocomposite as a substrate. AgNPs are dispersed on the surface of $Fe_3O_4$, then MIP will wrap the surface of the formed nanocomposite substrate. In this study the separation is also assisted magnetically. Polydopamine was used as a functional monomer as well as crosslinkers, while enrofloxacin was used as a template. The combination of $Fe_3O_4$/Ag is expected to increase the hotspot area during SERS analysis. Sui et al. [131] also developed analytical methods for antibiotic analysis with MIP SERS using AgESM MIP, utilizing eggshell as a support material while Li et al., used GO/Ag composite as a substrate combined with PVDF membrane as a support.

The results of research by Lu et al., showed that the detection of illegal biguanide derivatives in pharmaceutical preparations circulating in the trade could be done using the MIP-SERS combination analysis method. This study used graphene oxide (GO) nanocomposites with gold nanoparticles (AuNPs) as substrates. AuNPs were immobilized on the GO surface with the help of p-aminothiophenol. Next, the MIP was encapsulated on the surface of the nanocomposite. The template used is metformin. The results show that MIP@Au-GO SERS can analyze metformin HCl and phenformin HCl in the pharmaceutical preparation. This method is expected to be used to analyze and detect active drug compounds in complex matrix [132].

Two different researchers developed the analytical method for the detection of malachite green. The difference lies in the substrate composition and the type of MIP-SERS used. Zhang et al. used Au@AgNPs nanocomposite with single-stage MIP-SERS type [133], while Ekmen et al., used AgNPs as substrate with two-stage MIP SERS type. Ekmen et al., combined MIP with magnetic nanoparticles to increase the selectivity of the assay. Based on the study results, both methods can be used to detect malachite green on samples with complex matrix [134].

Liu et al., developed an Ag/GO/MIP sandwich nanostructure to analyze propranolol in complex matrix. In this structure, AgNPs are placed in the top position, and can interact directly with the target compound molecules to produce the best increase in SERS

signal. This method can be applied to other target molecules and is used to detect various pollutants with high sensitivity [135].

Zhou et al., developed molecularly imprinted polymer coated gold nanoparticles (MIP-AuNPs) as a material for detect and quantify L-Phenylalanine in one-step approach. Gold nanoparticles were prepared by bottom up method using sodium citrate as a reductor. The MIP-AuNPs hybrids were prepared by combination of sol gel method and molecular imprinting technology. MIP was in-situ formed on AuNPs by sol gel methods. L-phenylalanine was used as a template, TEOS as a crosslinkers and phenyltrimethoxysilane as a functional monomer. The MIP-AuNPs showed a good linearity and limit of detection. This material also can detect L-phenylalanine in the presence of its analogue, D-Phenylalanine and bovine serum [140].

The selective and sensitive MIP-SERS detection was also developed for determination of bisphenol A (BPA). Silver nanoparticles were synthesize by in-situ preparation inside molecularly imprinted polymer matrix for BPA detection. The MIP was prepared by using BPA as a template, EGDMA as a crosslinker, and AIBN as an inititator. Then amount of silver nitrate as a AgNPs precursor were added to the mixture. After the polymerization was complete, the rigid polymer was grounded. AgNPs were formed by add a reductor (Sodium borohydride) to the MIP powder. The recovery of this MIP-SERS method was calculated at 92.2% to 103.8%. According to spike sample, this method has a better recovery and relative standard deviation than HPLC methods. This method has a better limit of detection than HPLC methods [141].

The combination of magnetic core-shell SERS substrate was developed for antibiotics detection in water sample. $Fe_3O_4$@Ag composites was selected as a SERS substrate. In the end, the composites was then modified with dopamine to synthesize $Fe_3O_4$/Ag/MIP. $Fe_3O_4$ nanoparticles were synthesized by hydrothermal reaction of dopamine and and modified with amino. Enrofloxacin hydrochloride was chosen as a template. Dopamine was selected as a functional monomer and crosslinkers. This material was synthesize by pDA polymerization. This method has better performance and limit detection, compared to another method of synthesis [142].

The detection of trace-level antibiotic was developed by using a novel composite material, AgP/MIM. Ag/GO (graphene-oxide argentum) composite was used as a SERS substrate. To apply into practical sample detection, molecular imprinting technique was also introduced to improve the selectivity of the methods. Enrofloxacin hydrochloride was used as a template. AgNPs was synthesize by bottom up using silver nitrate as a precursor and ascorbic acid as a reductor. Ag/GO composite was synthesized by adding AgNPs into GO dispersion. The AgP/MIM was synthesize by step two precipitation polymerization at two different temperature. This method has better detection of limit and detection time, if we compared it to previous method [143].

Ag@MIPs hybrid was used in lysozyme analysis. Ag@MIPs hybrids was fabricated based on core-shell structure. Ag microsphere was synthesized by using ascorbic acid as a reductor. Ag microsphere and a mixture of MIP component were stirred. The lowest detection of limit concentration is 5 ng mL$^{-1}$. Ag@MIPs hydrid has better performance as a SERS substrate, comparing to Ag miscropshere itself. This method can develop into a promising detection method for biomolecules, pathogens and living cells [144]. Ag@MIP hybrid was also used for p-nitroaniline in aqueous environment. The results show that this material can be used as a SERS substrate. The obtained Ag@MIP exhibit good limit of detection, $10^{-12}$ M. Ag@MIP give better signal enhancement in SERS analysis than Ag nanoparticle itself [145]. Another application of MIP-SERS method is analysis of PAHs (polycyclic aromatic hydrocarbons). Au@MIPs was used as a SERS substrate. The Au@MIPs was fabricated by two-step procedure. Pyrene and fluoranthene, were used as a template. The combination of AuNPs and MIP solved each material's main limitation [146]. The detection of cloxacillin in pig serum were found to be more sensitive by using combination of MMIP (magnetic molecularly imprinted polymer) with SERS. The limit of detection was

7.8 pmol. The cloxacillin recoveries were found to be more 80%. This method can be used routinely to screen antibiotic residues in food products [147].

*4.2. Two Step MIP-SERS*

In two-stage MIP-SERS, MIP is separated from metal nanoparticles which are used as substrates in SERS analysis. Cao et al., analyzed the content of bitertanol, a triazide fungicide compound, in vegetable (cucumber) and fruit (peach) samples. MIP is used in this study to reduce interference from impurities that can interfere with the analysis results with SERS. In the manufacture of MIP, a dummy template is used, triadimefon, to prevent template leakage. It is expected to obtain better selectivity than using a template in the form of the analyte to be analyzed. MIP was packaged as a sorbent in an SPE cartridge during the analysis. The overall analysis time is 15 min. The developed method shows the same performance when compared to the previous methods (LC-MS, GC-MS, and HPLC-DAD), with a better analysis time [27].

Yan et al. also conducted a similar study to develop other triazine fungicide analysis methods, prometryn and simetryn, on rice and wheat samples. However, in this study, the same template was used with the analyte to be determined. This study's MIP-SERS analysis method was then compared with several previous analytical methods (GC-NPD, LC-DAD, MIP-SPE-HPLC, Fluorescence, AgNP-SERS, GC-TSD, CNT-Au-SERS). The results show that the newly developed Au-MIP-SERS method has the same performance as the previous method; even some parameters show better results [118].

The pesticide residue, chlorpyrifos, was also developed using the two-step MIP-SERS method. In this study, AgNPs were not only used as a substrate in SERS analysis but were also developed to be used as a colorimetric detection method for the same compound. The results showed that that method could be used to separate chlorpyrifos from apple juice samples. The AgNPs-colorimetric method could be used to detect chlorpyrifos in samples [136]. The analysis of herbicide residues (2,4-dichlorophenoxyacetic acid) in milk was developed by the two-step MIP-SERS method. MIP is then packaged in an SPE cartridge and was used to separate the residue of 2,4-dichlorophenocyacetic acid. AgNPs as a substrate for SERS were synthesized separately using sodium citrate as a reducing agent. The analysis time required is quite fast, only about 10 min. The resulting analytical method is sensitive for detecting residues in dairy products [137].

The combination of MIP-SPE and SERS was also analyzed for thiabendazole in apple juice. The analysis of thiabendazole was carried out in two stages. MIP is first synthesized and then used as an adsorbent on the SPE cartridge. AgNPs were synthesized from silver nitrate with trisodium citrate as a reducing agent.The total analysis time for thiabendazole analysis using MIP-SPE and SERS detection was 23 min. This analysis time is much faster than other traditional detection methods, which require complicated sample preparation [138]. The same analytical method was also developed to analyze atrazine in apple juice samples. This method uses AuNPs as substrate for detection with SERS. The results showed that this method can also be used to detect atrazine in other types of samples [139].

## 5. Conclusions and Future Prospective

Based on the explanation above, it can be concluded that the MIP-SERS is a potential method to be constantly developed for bioanalysis matrix. The analytical method developed is not limited to analyzing compounds in biological matrix (e.g., urine, serum, plasma), but also methods of trace analysis of compounds in agricultural product samples and residues in the environment.

Molecularly imprinted polymer can solve the limitation regarding SERS analysis. MIP and SERS in bioanalysis have a different function particularly in sample preparation stage. Combination of MIP and metal nanoparticles can resolve SERS drawback. The development of novel SERS substrate, either can be in the form of composite between MIP and metal nanoparticles, core-shell, or in another way of modification. Combination of MIP SERS can reduce analysis time and increase the detection limit.

The explanation show that the research excitement towards the discovery of new materials that can improve the performance of sample preparation in a series of analytical methods is tremendous. Based on research data obtained from mipdatabase.com, in 2021 and 2022, research papers investigate a lot about the combination of MIP with other forms of material to overcome the limitation in preparation of biological sample. Many studies have also led to the use of MIP as a method of detection and diagnosis.

The use of off-label drug for COVID-19 treatment during pandemic, causes bioanalytic methods indispensable. The bioanalytic method must be sensitive, selective, also accurate. The combination of MIP and SERS can overcome the difficulties that occur in a bioanalysis process, i.e., the sample preparation stage. Matrix complexity of bioanalytical sample can reduce the ability of method to obtain sensitive and accurate results.

Remdesivir, as one of drug of choice in the treatment of COVID-19, currently does not have an official standard analytical method established in any compendia. The analytical method's development is critical for clinical trials and for therapeutic drug monitoring. The most widely used analytical method is LC-MS or LC-MS/MS [148], using liquid-liquid extraction and/or solid-phase extraction as sample preparation methods. Developing the MIP-SERS-based analytical method is an excellent opportunity to obtain an excellent remdesivir analytical method for detection and quantification in biological preparations and samples. Cases of remdesivir preparations counterfeit also encourage the development of more sensitive and selective analytical methods.

**Author Contributions:** Conceptualization, H.A.W., S.I., R.R.M. and S.D.; methodology, H.A.W.; validation, S.I., R.R.M. and S.D.; resources, H.A.W., R.R.M.; writing—original draft preparation, H.A.W.; writing—review and editing, H.A.W. and S.D.; visualization, H.A.W.; supervision, S.I., R.R.M. and S.D. All authors have read and agreed to the published version of the manuscript.

**Funding:** Indonesia Endowment Fund for Education (LPDP RI) for doctoral scholarship and University Center of Excellence on Artificial Intelligence for Vision, Natural Languange Processing & Big Data Analysis (U-CoE AI-VLB), Institut Teknologi Bandung for the APC.

**Institutional Review Board Statement:** Not applicable.

**Informed Consent Statement:** Not applicable.

**Data Availability Statement:** Data sharing not applicable.

**Acknowledgments:** We would like to appreciate Indonesia Endowment Fund for Education (LPDP RI), School of Pharmacy ITB, and University Center of Excellence on Artificial Intelligence for Vision, Natural Languange Processing & Big Data Analysis (U-CoE AI-VLB), Institut Teknologi Bandung.

**Conflicts of Interest:** The authors declare no conflict of interest.

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
