# Peer review of "Molecularly-Imprinted SERS: A Potential Method for Bioanalysis"

_scipharm, doi:10.3390/scipharm90030054_

Round 1

Reviewer 1 Report

This paper introduced the joint analytical method of SERS and MIP. In general, the language, the logics, and the entire flow need further polishing and references are not adequate while the paper presents certain contribution to the knowledge body.

1. Please clear the inappropriate hyphens like ‘ana-lytes’, ‘quant-tative’, ‘manu-facturing’, ‘durablil-ity’,’im-printing’ in page 2. There are too many to list out. Please double check the whole paper.

2. There are many language issues causing confusion.

2.1 The title is ‘A molecularly-imprinted SERS for analytical method’. ‘A molecularly-imprinted SERS’ itself is an analytical method, so why it is ‘for analytical method’. Method is countable.

2.2 In the abstract,

‘In developing bioanalytical methods, the most challenging step is the presence of complex 11 matrices. Therefore, the sample preparation method plays an essential role in increasing the selectivity and sensitivity of bioanalysis.’ Why complex matrcies is challenging so the sample prep is essential in increasing the performance???

Line 16, MIP and SERS combination is widely developed for bioanalysis, but why in line 23 it is still expected to assist bioanalytical methods? The logic is not clear here.

2.3 The introduction definitely needs reconstruction.

In the first paragraph, the authors claimed nanotech application fields, and limits. The second paragraph claimed nanotech has been applied in some fields, again.

Line 42, ‘The recent applications in analytical science are especially for monitoring 41 production processes, characterization, and use of end products. Therefore, nanomaterials are increasingly being used in analytical processes.’ Why recent applications in these fields so the nanomaterials are increasingly being used???

Line 46, ‘The sample preparation stage in bioanalysis is crucial because of its matrix complexity’. Why….

Line 47, the paragraph just ends with ‘The sample's preparation is followed by qualitative and quantitative analysis with specific instruments, such as HPLC, LC-MS, SERS, LC-MS/MS, and 48 many more.’ The next paragraph just jumped to sample prep and has nothing with these instrumental methods. So what’s the purpose of this sentence? The authors want to compare SERS against other methods? Or just simply listed all the possible methods but regardless of non-SERS methods or sth? This full stop is a little confusing.

Line 51, SPE needs abbreviation.

Line 65, ‘SERS (surface-enhanced Raman spectroscopy) is a detection technique that combines Raman spectroscopy with nanotechnology. As a result of massive boosting effects, even a 66 single molecule may be identified.’ The readers will expect a concise but clear definition in one or multiple sentences.  This sentence definitely did not define SERS sufficiently while the next few sentences show nothing related to the definition. Raman plus nanotech is absolutely not equal to SERS. The paper is about MIP and SERS. If SERS itself is not defined clearly, how could this paper help the readers?

Line 79, ‘As a result, finding novel materials for analyte identification is crucial in order to widen the applications of SERS in complex matrixes (wrong spelling BTW)’. What’s the relationship between finding novel materials to the applications??? Novelty does not necessarily mean better than the existing materials. The logic needs to refine.

There are still many issues like above, please double check and make sure all the sentences are in correct logic flow.

3) Citations are absolutely not enough either in the introduction or the main body.

For example, the entire part 2.4 has no citations at all. If the authors feel this part is already comment sense and then no need to waste a whole page to explain, otherwise, citations are required.

Line 66, the authors claimed a single molecule may be identified but no citation.

I noticed the authors citied 114 papers, which is an acceptable amount for review papers, while the text is a little short compared to the similar review and some sections have no citations at all. Here is my suggestion. That’s great to get a big number of existing works visualized; however, instead of mixing together as many as possible, it’s better to framework and categorize a little more with less papers, then deeply understand their pros and cons, novelties and limitations. A graphic summary is also recommended, which can demonstrate the authors’ own thoughts. A good review could be considered as an educational material to readers who are not familiar with this topic or want to dive deep in this field. Therefore, a clear flow and overarching comparison and summary would be really helpful.

4) Last but not least, I read through these two reviews published in 2020. I feel they have clearer framework, smooth flow, and the comprehensive scope. Can the authors help me understand what are the advantages of the current paper compared with these two?

https://doi.org/10.1016/j.talanta.2020.122031

https://doi.org/10.1021/acssensors.9b02039

Reviewer 2 Report

In this manuscript, the authors review a novel analytical method of MIP SERS, which is the combination of MIP and SERS. In my opinion, the review is helpful to understand the application of MPI in SERS for the bioanalysis. Therefore, I would like to recommend the publication of the review in Scientia Pharmaceutica after minor revisions. Detail comments are as follows.

1.      Title. I would recommend the following possible title: Molecularly-imprinted SERS: a powerful analytical method.

2.      Introduction. Line 81-85. This paragraph focuses on the SERS, so description on MIP is not necessary.

3.      Line 60. One work (Analytical Chemistry, 90(20): 11827-11834, 2018.) is suggested to be added in Reference [3].

4.      Line 92. A recent work (Biosensors and Bioelectronics,142, 111533, 2019.) should be added in Reference [8],[13],[14].

5.      Some reference numbers in Table 1 are missing in the main text. For example, Ref. 111.

6.      In Fig. 1, the structures and names of all chemicals are sugggested to be redrawn or rewritten for clarity.

7.      In Table 1. What is the criterial for the sequence of the examples? Please try to make the table more brief.

8.      Some references are recited in the Refereces, for example, Ref. 80 and 86. Please check!

9.      More figures are encouraged to be added in the manuscript.

10.   Language in the manuscript must be improved.

Round 2

Reviewer 1 Report

The revision is satisfying.